# Late Pleistocene Boulder Slumps Eroded from a Basalt Shoreline at El Confital Beach on Gran Canaria (Canary Islands, Spain)

Inés Galindo [1], Markes E. Johnson [2,*], Esther Martín-González [3], Carmen Romero [4], Juana Vegas [1], Carlos S. Melo [5,6,7,8], Sérgio P. Ávila [6,8,9] and Nieves Sánchez [1]

1   Instituto Geológico y Minero de España, Unidad Territorial de Canarias, c/Alonso Alvarado, 43, 2A, 35003 Las Palmas de Gran Canaria, Spain; i.galindo@igme.es (I.G.); j.vegas@igme.es (J.V.); n.sanchez@igme.es (N.S.)
2   Department of Geosciences, Williams College, Williamstown, MA 01267, USA
3   Museo de Ciencias Naturales de Tenerife, Organismo Autónomo de Museos y Centros, C/ Fuente Morales, 1, 38003 Santa Cruz de Tenerife, Spain; MMARTIN@museosdetenerife.org
4   Departamento de Geografía, Campus de Guajara, Universidad de La Laguna, La Laguna, 38071 Tenerife, Spain; mcromero@ull.esdu.es
5   Departamento de Geologia, Faculdade de Ciências, Universidade de Lisboa, 1749-016 Lisbon, Portugal; csmelo@fc.ul.pt
6   CIBIO–Centro de Investigação em Biodiversidade e Recursos Genéticos, InBIO Laboratório Associado, Pólo dos Açores, Universidade dos Açores, Rua da Mãe de Deus, 9500-321 Ponta Delgada, Portugal; sergio.pa.marques@uac.pt
7   IDL–Instituto Dom Luiz, Faculdade de Ciências, Universidade de Lisboa, 1749-016 Lisbon, Portugal
8   MPB–Marine Palaeontology and Biogeography lab, Universidade dos Açores, Rua da Mãe de Deus, 9500-321 Ponta Delgada, Portugal
9   Faculdade de Ciências da Universidade do Porto, Rua do Campo Alegre 1021/1055, 4169-007 Porto, Portugal
*   Correspondence: markes.e.johnson@williams.edu; Tel.: +1-413-2329

**Abstract:** This study examines the role of North Atlantic storms degrading a Late Pleistocene rocky shoreline formed by basaltic rocks overlying hyaloclastite rocks on a small volcanic peninsula connected to Gran Canaria in the central region of the Canary Archipelago. A conglomerate dominated by large, ellipsoidal to angular boulders eroded from an adjacent basalt flow was canvassed at six stations distributed along 800 m of the modern shore at El Confital, on the outskirts of Las Palmas de Gran Canaria. A total of 166 individual basalt cobbles and boulders were systematically measured in three dimensions, providing the database for analyses of variations in clast shape and size. The goal of this study was to apply mathematical equations elaborated after Nott (2003) and subsequent refinements in order to estimate individual wave heights necessary to lift basalt blocks from the layered and joint-bound sea cliffs at El Confital. On average, wave heights in the order of 4.2 to 4.5 m are calculated as having impacted the Late Pleistocene rocky coastline at El Confital, although the largest boulders in excess of 2 m in diameter would have required larger waves for extraction. A review of the fossil marine biota associated with the boulder beds confirms a littoral to very shallow water setting correlated in time with Marine Isotope Stage 5e (Eemian Stage) approximately 125,000 years ago. The historical record of major storms in the regions of the Canary and Azorean islands indicates that events of hurricane strength were likely to have struck El Confital in earlier times. Due to its high scientific value, the outcrop area featured in this study is included in the Spanish Inventory of Geosites and must be properly protected and managed to ensure conservation against the impact of climate change foreseen in coming years.

**Keywords:** coastal storm deposits; storm surge; hydrodynamic equations; upper pleistocene; marine isotope substage 5e; North Atlantic Ocean

## 1. Introduction

Evidence for the influence of hurricanes in the northeast Atlantic Ocean during the Late Pleistocene is based on analyses of storm beds preserved on Santa Maria Island in the Azores archipelago [1] and Sal Island in the Cabo Verde archipelago [2]. This line of research follows a growing interest in coastal geomorphology as related to the accumulation of mega-boulders attributed to superstorms or possible tsunami events [3]. Documentation of eroded mega-boulders from rocky shores in Mexico's Gulf of California during the subsequent Holocene [4–6] adds to our knowledge of the physical scope of deposits that can be directly related to historical storm patterns. The Canary Islands in Spain represent seven main islands and numerous islets and seamounts in the North Atlantic located over lithospheric fractures off the northwest coast of Africa, with radiometric dates that vary from 142 Ma to 0.2 Ma along a general east to west axis [7]. At the center of the archipelago, Gran Canaria Island has a volcanic history dominated by subaerial development dating back to 13.7 Ma with successive stages of construction and erosion of the island edifice [8].

The attraction of Gran Canaria for this study is based on a distinctive rocky paleoshore formed by a 3-m thick lava flow exposed laterally along El Confital beach, on the southwest side of La Isleta peninsula, located in the northeastern quarter of Gran Canaria Island. Subaerial flows around several craters preserved atop the Isleta volcanoes exhibit variable Pliocene to Quaternary ages [8], but the basalt shore at El Confital beach is dated to approximately 1 Ma [9,10]. Prior to this contribution, a paleontological study at El Confital beach described an older assemblage of Miocene fossils at one end and a more extensive assemblage of intertidal to shallow subtidal invertebrates confined to the last interglacial epoch attributed to the Eemian Stage, also correlated with Marine Isotope Substage 5e, approximately 125,000 years in age [11]. The goals of this subsequent study include a review of the marine fossil fauna with particular emphasis on species zonation, and with an added emphasis on the extent of encrustations by coralline red algae. Most importantly, this work features extensive analyses of boulders eroded from the paleoshore, among which the Upper Pleistocene biota is preserved. In particular, the physical analyses conducted for clast shape and size are integral to estimates of wave heights based on competing mathematical equations applied to a rocky shoreline composed of joint-bound basalt and hyaloclastite layers.

The range in estimated wave heights extrapolated from average boulder size, contrasted against maximum boulder size at multiple sample sites, suggests that a pattern of repetitious storms is implicated with the coastal erosion of La Isleta and more generally of Gran Canaria Island over an extended period of Late Pleistocene time. The viability of this inference is tested against the historical record of cyclonic disturbances in the eastern North Atlantic Ocean around the Azorean and Canary Islands. Increasing interest in the historical record of storms in this region is available mainly in Spanish-language reports, but also well summarized in the international literature [12]. This approach with reference to historic storms is predicated on a similar analysis of coastal storm beds from the Pleistocene of Santa Maria Island in the Azores [1] as well as coastal boulder beds from the Holocene of Mexico's Baja California [4–6].

## 2. Geographical and Geological Setting

Gran Canaria is the third largest and most centrally located island in the Canary Archipelago, situated 150 km off the northwest coast of Africa (Figure 1a,b). With an area of 1560 km², the island exhibits a circular map outline with an outer perimeter of about 50 km showing a pattern of ravines radiating from the island center at an elevation of 1949 m above present sea level. La Isleta peninsula lies in the northeastern part of Gran Canaria Island, now linked to it by a sandy isthmus about 200 m in width and around 2 km long (Figure 1b,c). Gran Canaria island emerged during the Miocene and reflects a complex geological evolution including the formation and erosion of several stratovolcanoes. Pliocene-Quaternary mafic fissure eruptions were concentrated along the northeastern half of the island. Volcanic activity on La Isleta began during the early

Pleistocene with the formation of submarine volcanoes more than 1 Ma ago [9,10]. At least two submarine Surtseyan eruptions separated by marine sedimentary rocks have been identified. The later subaerial phase includes several mafic fissure eruptions (Figure 1c). In El Confital Bay, located off the east-southeast part of La Isleta, erosion has dominated since the middle Pleistocene (>152 ka) and marine, aeolian, and colluvial deposits have been deposited overlying the volcanic sequence (Figure 1d). This area was intensively modified by quarry operation, military activities, and expansion of a shantytown. The clearing of the coastal zone after illegal settlement between 1960 and 1995 and its subsequent elimination by civil authorities, resulted in the area of the deposit encompassed by a larger part of the Confital platform being compromised.

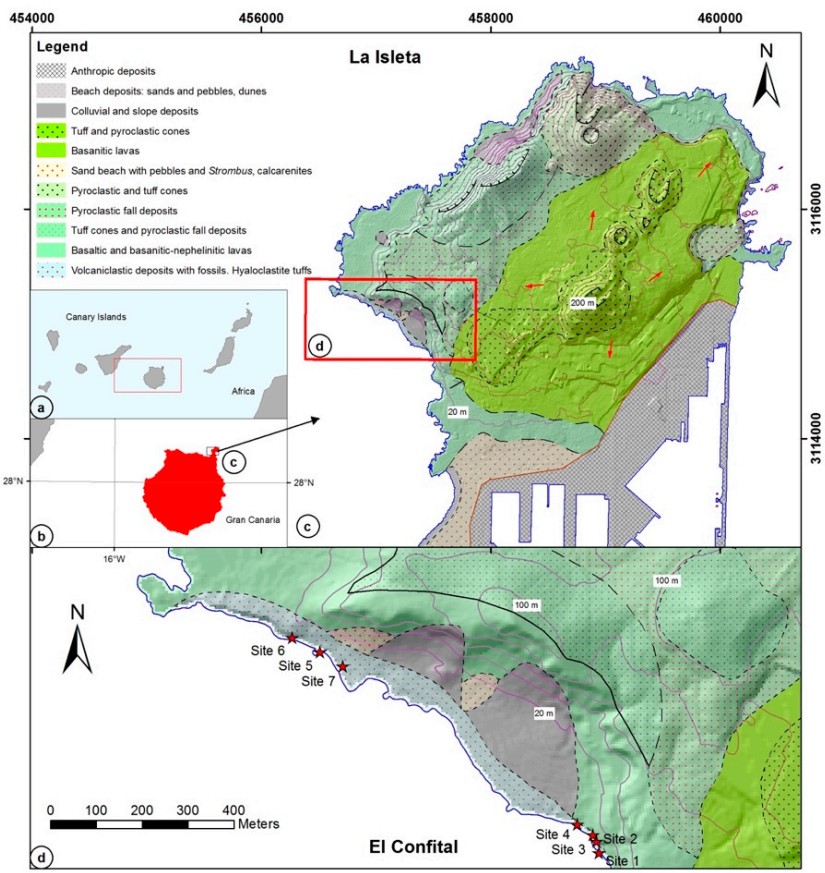

**Figure 1.** Canary Archipelago and features of the centrally located Gran Canaria Island: (**a**) Canary Archipelago with inset showing the position of Gran Canaria; (**b**) central island of Gran Canaria with an arrow pointing to La Isleta on the northern periphery of the island; (**c**) geological map of the volcanic edifice that forms La Isleta with inset box marking El Confital beach (coordinates in UTM m, H28 R); (**d**) map of the kilometer-long beach at El Confital showing seven localities from which sedimentological and paleontological data were drawn for this study. Modified from the 1990 Geological Map by Balcells and Barrera at a scale of 1:25,000 [10].

El Confital bay has a U-shape open to the southwest and is bound on the northeast by an escarpment more than 100 m high. The beach has been developed over a littoral platform (Figure 1c,d) covering approximately 0.1 km$^2$ that consists of highly fractured hyaloclastite tuffs from the submarine stage and volcanoclastic deposits including marine fossils. Parallel to the beach, a basalt cliff line extends parallel to the coast for a broken distance of about a kilometer. Cliffs crop out at an elevation of 1 m in the central part of the area and rise to more than 10 m at opposite ends in the southeast and northwest. In these parts, tuffs are overlain by subaerial lava flows and the cliff reaches a height of more than 10 m.

## 3. Materials and Methods

### 3.1. Data Collection

Gran Canaria was visited in April 2018, when the organizing author was invited to appraise the beach and rocky shoreline at El Confital beach on the east side of Las Palmas de Gran Canaria. The original data for this study was collected in October 2020 from deposits dominated by basalt boulders consolidated within a limestone matrix. Individual basalt clasts from six stations were measured manually to the nearest half centimeter in three dimensions perpendicular to one another (long, intermediate, and short axes). Differentiated from cobbles, the base definition for a boulder adapted in this exercise is that of Wentworth [13] for an erosional clast equal to or greater than 256 mm in diameter. No upper limit for this category is defined in the geological literature [14].

Triangular plots are employed to show variations in clast shape, following the design of Sneed and Folk [15] for river pebbles. In the field, all measured clasts were characterized as sub-rounded and a smoothing factor of 20% was applied uniformity to adjust for the estimated volume first calculated by the simple multiplication of the lengths of the three axes. Comparative data on maximum cobble and boulder dimensions were fitted to bar graphs to show size variations in the long and short axes from one sample to the next. Comparative data on maximum cobble and boulder dimensions were fitted to bar graphs to show size variations in the long and intermediate axes from one sample to the next. The rock density of basalt from the Pleistocene sea cliffs on El Confital beach is based on laboratory analyses in an unpublished PhD thesis that yields a value of 2.84 g/cm$^3$ [16].

### 3.2. Hydraulic Model

Dependent on the calculation of rock density for basalt, a hydraulic model may be applied to predict the force needed to remove cobbles and boulders from a rocky shoreline with joint-bound blocks as a function of wave impact. Basalt is the typical extrusive volcanic rock characteristic of many oceanic islands. Herein, two formulas are applied to estimate the size of storm waves against joint-bounded blocks derived, respectively from Equation (36) in the work of Nott [17] and from an alternative formula using the velocity equations of Nandasena et al. [18], as applied by Pepe et al. [19].

$$Hs = \frac{\left(\frac{\rho_s - \rho_w}{\rho_w}\right)a}{C_1} \tag{1}$$

$$u^2 = \frac{2\left(\frac{\rho_s - \rho_w}{\rho_w}\right)g\,c\,(\cos\theta + \mu_s \sin\theta)}{C_1} \tag{2}$$

where $Hs$ = height of the storm wave at breaking point; $\rho_s$ = density of the boulder (tons/m$^3$ or g/cm$^3$); $\rho_w$ = density of water at 1.02 g/cm$^3$; a = length of the boulder on long axis in cm; $\theta$ is the angle of the bed slope at the pre-transport location (1° for joint-bounded blocks); $\mu_s$ is the coefficient of static friction (=0.7); $C_1$ is the lift coefficient (=0.178). Equation (1) is more sensitive to the length of a boulder on the long axis, whereas Equation (2) is more sensitive to the length of a boulder on the short axis. Therefore, some differences are expected in the estimates of $H_S$.

## 4. Results

### 4.1. Characteristics of the Conglomerate

Two principal facies of Pleistocene conglomerates occur at El Confital. The first is linked to the background cliffs at opposite ends of the bay and the second is related to the open paleo-platform 5 to 10 m distal from the cliffs in the more central part of the beach. Along the higher cliffs, a marine conglomerate fills paleo-channels and forms ridges trending NNE–SSW perpendicular to the shore (Figure 2a,b). The channel conglomerate is well cemented and consists of coarse-grained (pebble to boulder size) clasts embedded in a white matrix of bioclastic calcarenite. The matrix incorporates marine mollusks

and calcareous algae. Commonly reaching a cubic meter in size, boulders are generally ellipsoidal to angular in shape and well rounded. The largest boulders exhibit a long axis in excess of 2 m. Locally, the matrix is composed of reddish sands derived from eroded hyaloclastite tuffs (Figure 2c). Neptunian dykes related to the emplacement of this deposit cut into the tuffs and volcanoclastic deposits filled with carbonates and zeolites as well as bioclasts and gravels (Figure 2d). The more distal conglomerate is deposited nearly horizontal across the shore platform. Here, boulders are smaller and more rounded, and the matrix consists of coarse-grained bioclastic sand.

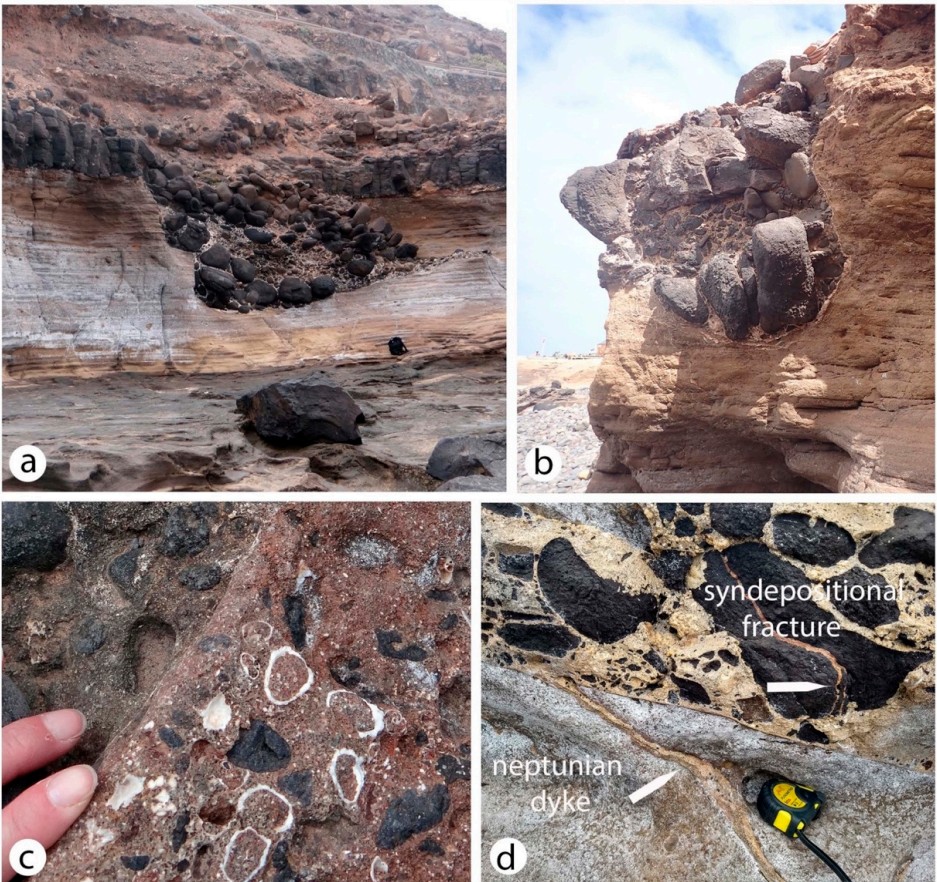

**Figure 2.** Conglomerate slumps above El Confital beach south of La Isleta; (**a**) incision approximately 3.5 m deep in cliff face filled with basal boulders loosened from the adjacent basalt formation (backpack for scale); (**b**) boulder-filled cut in the same cliff face with individual boulders exposed in bas relief (the largest boulder at the bottom has a long axis of approximately 75 cm); (**c**) reddish sands derived from eroded hyaloclastite; (**d**) details showing filling of a neptunian dyke and syndepositional fractures in basalt (tape-measure case for scale).

*4.2. Comparative Variation in Clast Shapes*

Raw data on clast size in three dimensions collected from each of six sample sites are recorded in Appendix A (Tables A1–A6). Regarding shape, points representing individual cobbles and boulders are fitted to a set of Sneed-Folk triangular diagrams (Figure 3a–f). The spread of points across these plots reflects a consistent pattern in the variation of shapes from one sample to another. As few as one to three points fall within the upper triangle in each diagram, which represents an origin from a perfectly cube-shaped endpoint as a joint-bound block of basalt. No more than five points from each sample fall within the lower, right-hand rhomboid in these diagrams. This extreme corner signifies elongated blocks with one super-attenuated axis in relation to two axes that are significantly shorter by 75% or more. The result is a bar-shaped piece that originated as a joint-bound block of basalt. The plot with the greatest number of points in these two extremes is from locality 5 (Figure

3e), from a position closer to the western end of the paleoshore. Numbering between 10 and 15 per sample, the majority of points from each of the six samples fall within the central two rhomboids directly below the top triangle. Such points clustered at the core of a Sneed-Fold diagram are typical of clasts for which two of the three dimensions are closer in value than the length of the third axis. It is notable that no points appear anywhere along the margin of rhomboids on the left side of these diagrams. It is clear that no tendency in shape towards plate-shaped clasts is evident in the data. The general slope of points in agreement from the six plots follows a uniformly diagonal trend from the top to the lower right-hand corner. The trend in distributed points from these plots signifies the rounding of clasts in which two of the dimensions (maximum and intermediate lengths) are more closely matched with the third as an outlier.

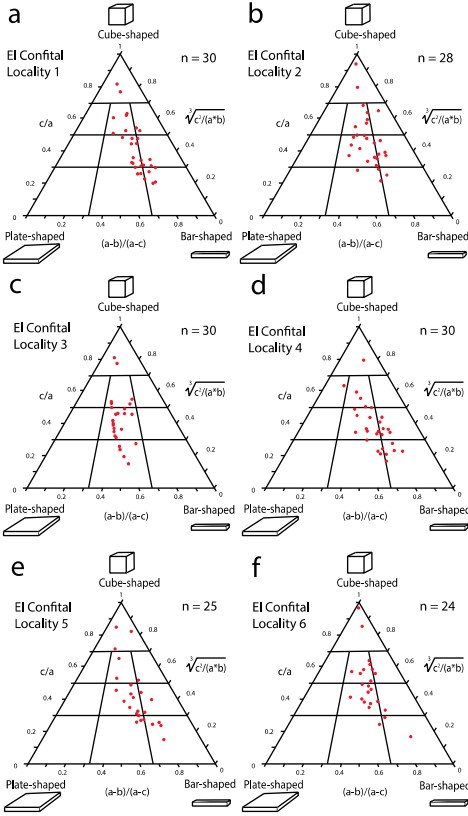

**Figure 3.** Set of triangular Sneed-Folk diagrams used to appraise variations in cobble and boulder shapes sampled along the Upper Pleistocene paleoshore east to west at El Confital beach south of La Isleta: (**a**–**f**) sample localities 1 to 6 represented, respectively.

### 4.3. Comparative Variation in Clast Sizes

Drawn from original data (Tables A1–A6), clast size is plotted to the best effect on bar graphs as a function of frequency against maximum and intermediate lengths of the two longest axes perpendicular to one another. The dozen graphs plotted (Figure 4a–l) exhibit trends in clast size sorted by intervals of 15-cm, in which the boundary between cobbles and boulders is embedded within the range for clasts between 16 and 30 cm in diameter. The left-hand column (Figure 4a,c,e,g,k) depicts lateral variations in maximum boulder length from the six samples on an east to west transect along the Upper Pleistocene paleoshore. Overall, each of the samples in this dimension is numerically dominated by boulders in contrast to cobbles at ratios from 3:2 and 4:1. Samples from the east end (Figure 4a,c) reflect differences that conform to a normal bell-shaped curve, whereas samples from the west end (Figure 4i,k) are strongly skewed to include a few boulders of extreme size in excess of one meter. In contrast, the right-hand column (Figure 5b,d,f,h,j,l) shows lateral variations in values for intermediate length of clasts. In all but two examples (Figure 4f,j) the number

of measurements falling within the interval of 16 to 30 cm exceeds those registered for the same interval as measured for the long axis. This result reflects a general shift in size to smaller frequencies compared to the left-hand column and confirms the diagonal trend in shapes illustrated by the Sneed-Folk diagrams (Figure 4a–f). Skewness that mirrors the inclusion of extra-large clasts is especially evident in the bar graphs at the extreme ends of the paleoshore (Figure 4b,l). The largest basalt clast identified in the entire project registered a long axis of 214 cm, an intermediate axis of 94 cm, and short axis of 50 cm (Table A6).

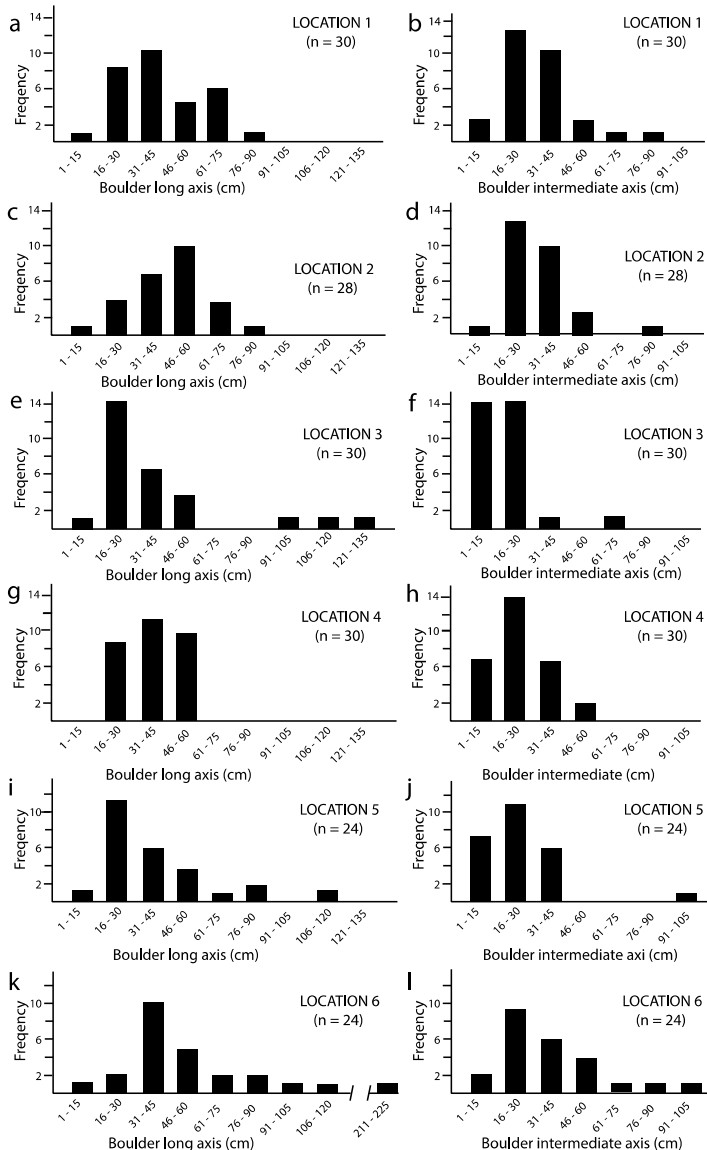

**Figure 4.** Set of bar graphs used to contrast variations in maximum and intermediate boulder axes from six samples at El Confital beach south of La Isleta: (**a**,**b**) bar graphs from locality 1; (**c**,**d**) bar graphs from locality 2; (**e**,**f**) bar graphs from locality 3; (**g**,**h**) bar graphs from locality 4; (**i**,**j**) bar graphs from locality 5; (**k**,**l**) bar graphs from locality 6.

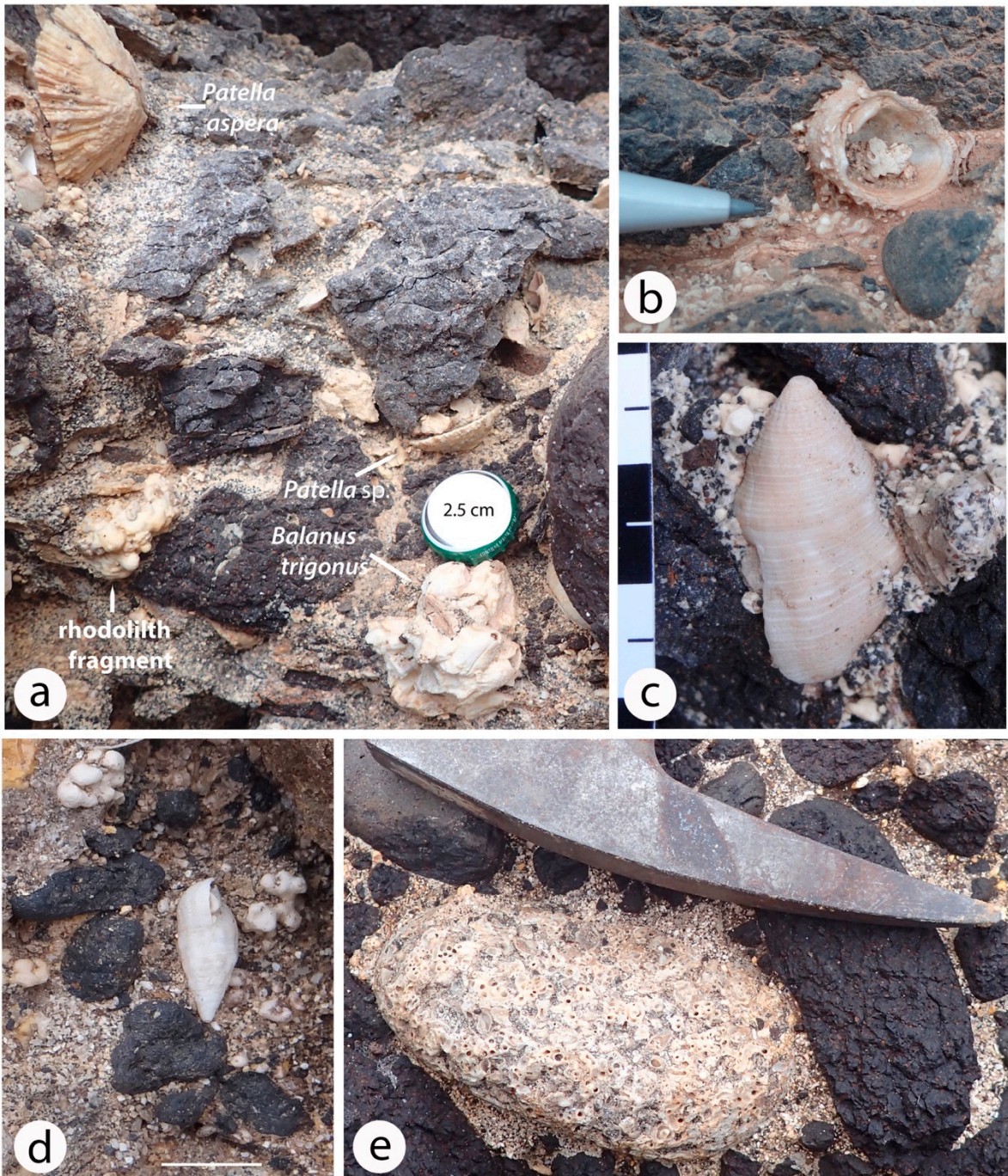

**Figure 5.** Examples of abundant marine fossils from the paleoshore exposed at El Confital beach: (**a**) filling among larger boulders that includes the elbow shell (*Patella aspera*) and barnacles (*Balanus trigonus*) with bottlecap for scale; (**b**) bivalve (*Chama gryphoides*) with pen-tip for scale; (**c**) gastropod (*Gemophos viverratus*) with 3-cm scale; (**d**) gastropod (*Cerithium vulgatum*) with scale bar = 1 cm; (**e**) large colony of gastropods (*Dendropoma cristatum*) with rock hammer for scale.

### 4.4. Paleontological Inferences on Water Depth

A moderately rich molluscan fauna of 42 marine gastropods and eight bivalves from El Confital (Table 1) that also includes barnacles and rhodoliths formed by coralline red algae is maintained in the permanent collections of the Tenerife Science Museum. Overall, these fossils reflect organisms that lived under inter-tidal to shallow subtidal conditions as confirmed by outcrop relationships, where currently the fossil remains are visible cemented in place along small sections of the bay and in ravines that form after heavy rains.

The Pleistocene fauna at El Confital corresponds to a high-energy setting against a rocky shore highlighted by the abundance of mäerl and rhodoliths associated with relatively abundant patelid gastropods (Figure 5a), chamid bivalves (Figure 5b), and other mollusks (Figure 5c,d). Extensive colonies of vermetid gastropods (*Dendropoma cristatum*) are represented as discrete biological clasts incorporated within the conglomerate (Figure 5e). Although extensive shell fragmentation is evident, it is worth noting the high rate of complete shells, which include delicate ornamentation. In the case of pateliform shells, preservation also features evidence for stacking. This taphonomic trait is characteristic of a high-energy regime. At the base of the deposit on the platform at El Confital (Figure 2, locality 7), there is evidence of bioturbation possibly related to the activity of crabs. Also, trace fossils likely related to the activity of polychaets are preserved within the neptunian dykes that are common on the platform.

**Table 1.** Summary species list of epifaunal, infra-intertidal invertebrates from the Upper Pleistocene strata exposed at El Confital beach correlated with Marine Isotope Substage 5e. Extinction is denoted by asterisk.

| Phylum | Class | Species | Phylum | Class | Species |
|---|---|---|---|---|---|
| Mollusca | Gastropoda | *Acanthina dontelei* * <br> *Alvania macandrewi* <br> *A. scabra* <br> *Barleiia unifasciata* <br> *Bolma rugosa* <br> *Bittium reticulatum* <br> *Bursa scrobilator* <br> *Cerithium vulgatum* <br> *Cheilea equestris* <br> *Clanculus berthelotii* <br> *Columbella adansoni* <br> *Conus guanche* <br> *C. pulcher* <br> *Coralliophila meyendorffi* <br> *Dendropoma cristatum* <br> *Diodora gibberula* <br> *Erosaria spurca* <br> *Gemophos viverratus* <br> *Gibbula candei* <br> *Gibbula* sp. <br> *Haliotis tubeculata* <br> *Littorina littorea* <br> *Luria lurida* <br> *Manzonia crassa* <br> *Marginella glabella* <br> *Mitra cornea* <br> *Monoplex parthenopeus* <br> *Naria spurca* <br> *Patella aspera* <br> *P. candei* <br> *P. crenata* <br> *P. piperata* <br> *Phorcus atratus* <br> *P. sauciatus* <br> *Pusia zebrina* <br> *Stramanita haemastoma* <br> *Tectarius striatus* <br> *Thylacodes arenarius* <br> *Vermetus triquetrus* <br> *Vermetus* sp. <br> *Vexillum zebrinum* <br> *Zebina vitrea* | Mollusca | Bivalvia | *Barbatia barbata* <br> *Bractechlamys corallinoides* <br> *Cardita calyculata* <br> *Chama gryphoides* <br> *Ctema decussata* <br> *Glycymeris glycymeris* <br><br> *Pecten* sp. <br> *Venus verrucosa* |
| | | | Arthropoda | Cirripedia | *Balanus trigonus* |

In addition to preservation of whole but also fragmented rhodolith debris (Figure 6a,d) that signify the remains of a Pleistocene mäerl bed at El Confital, deposits are notable for the conglomerate consisting of mixed basalt cobbles and large boulders that exhibit erosional smoothing. The growth of coralline red algae in thin layers encrusted around and among these basalt clasts is widespread (Figure 6). In some examples, algal crusts are localized

and fail to completely surround individual cobbles leaving some parts free as viewed profile (Figure 6a). This scenario implies that some clasts were only partially coated by crustose algae elsewhere and subsequently were transferred to the conglomerate. In other examples, thick growth of algal crusts completely fills voids between boulders and small cobbles fixed in between (Figure 6b). This suggests an alternative scenario in which algal growth occurred perhaps on the outer margin of the conglomerate after its mass accumulation. Crustose red algae also are found in patches attached to basalt boulders exposed in three-dimensional relief (Figure 6c).

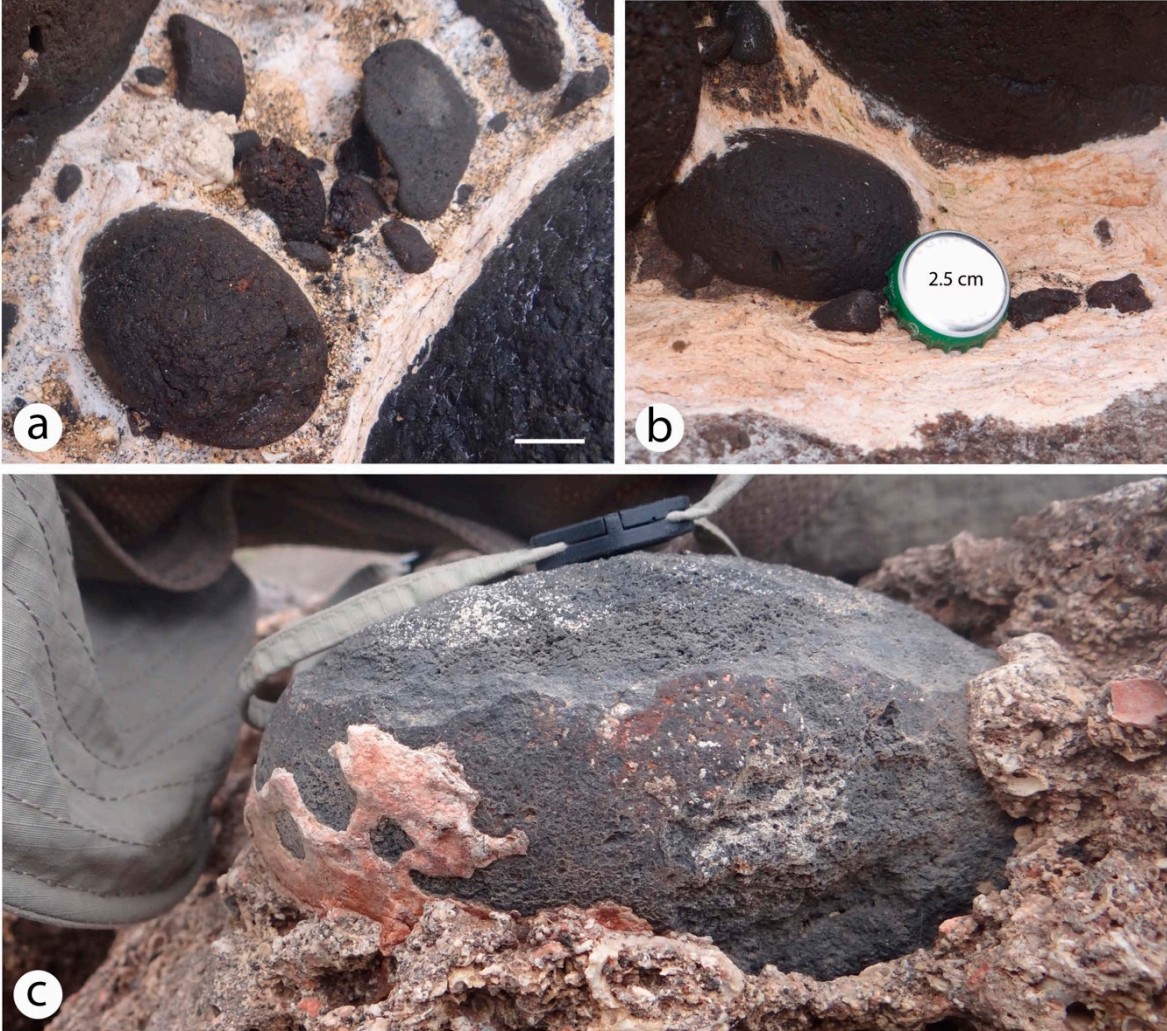

**Figure 6.** Examples of basalt cobbles and boulders heavily encrusted by coralline red algae: (**a**) space between two adjacent boulders filled by algal-encrusted cobbles and pebbles (scale bar = 2.5 cm); (**b**) close-up view of thick algal encrustations in the space among basalt clasts (bottle cap for scale); (**c**) hat-size basalt boulder with remnants of coralline red algae (clasp on hat string = 3 cm).

### 4.5. Storm Intensity as a Function of Estimated Wave Height

Clast sizes and maximum boulder volumes drawn from the six field localities are summarized in Table 1, allowing for direct comparison of average values for all clasts, as well as values for the largest clasts in each sample based on Equations (1) and (2) derived from the work of Nott [16] and Pepe et al. [19].

The Nott formula [17] shown in Equation (1) yields an average wave height of 4.5 m for the extraction of joint-bound blocks from basalt sea cliffs exposed at El Confital beach and their subsequent transfer as slump boulders in samples 1 to 6. A much larger value

for a wave height of 11 m is derived from the average of the largest single blocks of basalt recorded from the six locations. More sensitive to clast length from the short axis, the more sophisticated Equation (2) from Pepe et al. [19] yields values that are consistently lower for the estimated average wave height and estimated maximum wave height. The difference between the two calculations for estimated average wave height as well as estimated maximum wave height is, however, very small at 30 cm. Notably, the value for average maximum wave height derived from Equation (1) from Nott [17] is 2.5 times as high as the value found for overall average wave height using the same formula. Essentially the same factor applies to the slightly lesser values derived from Equation (2) according to Pepe et al. [19]. Clearly, the hydraulic pressure with extreme wave impact is necessary to loosen and budge the largest fault-bound blocks of basalt in the Pleistocene cliff line, as represented by the enormous block at locality 6 estimated to weigh 2.25 metric tons (Table A6). The essential issues under consideration in the following sections pertain to the singularity of a lone event of extreme magnitude as opposed to the repetition of many, but less energetic events in the shaping of the Pleistocene boulder slumps at El Confital beach.

## 5. Discussion

### 5.1. Integration of Paleontological and Physical Data

Boulders derived from horizontal layers of joint-bound basalt that originated as a subaerial flow about 1 million years ago at El Confital are estimated to have undergone wear that resulted in a 20% reduction in volume from more cubic or bar-shaped blocks (Figure 4) due to mutual friction under wave shock and subsequent erosion that smoothed sharp corners. Pleistocene fossils incorporated within the resulting conglomerate (Table 1, Figure 5) reflect an age correlated with Marine Isotope Stage 5e during the last interglacial epoch [11,20–23].

The assemblage represents a high-diversity, intertidal to shallow subtidal fauna dominated by mollusks that thrived on a basalt shelf on the southern margin of a small volcanic edifice. Except for those few limited to the MIS5e, most of the species listed in Table 1 continue to inhabit the contemporary coasts of the Canary Islands. These taxa are characterized by a preference for rocky or mixed littoral bottoms (sandy with rocky clasts of different sizes) up to a depth of about 3 m within the intertidal zone. The various species occupy different niches, such as crevices or intertidal pools or beneath rocks, which provide protection from the intense northwesterly waves that dominate the shores of the Canary Islands. In the case of El Confital today, there also occurs a wide rocky intertidal flat with little slope. In these shallows, the organisms are grouped in parallel bands or remain associated with pools at low tide, depending on their ability to adapt to environmental factors such as desiccation, temperature, salinity, and water agitation that condition life in that environment [23]. The intertidal shallows are home to a high number of marine organisms, highlighted by the dominant populations of pateliform and trochid gastropods.

Some of the larger clasts and boulders are encrusted with crustose red algae exhibiting rinds in excess of a centimeter in thickness (Figure 6). In places, surviving patches of red algae retain a rose coloration typical of coralline red algae (Figure 6c), but it may be due to inorganic discoloration. In addition to platy red algae cemented directly onto cobbles and boulders, whole rhodoliths and the debris of broken rhodoliths occur in pockets scattered throughout the conglomerate (Figure 5a). Some faunal elements may have lived within the interstices of adjacent boulders after the conglomerate was formed during slump events. Platy red algae are concentrated unevenly on the sides of cobbles and smaller boulders (Figure 6a), leaving other faces vacant. Some open spaces between adjacent boulders appear to have been filled by the continual growth of crustose red algae (Figure 6b). On the other hand, rhodoliths potentially composed of the same species of coralline red algae in unattached growth forms expressed by spherical shapes would have expired due to a lack of mobility.

A proper survey has yet to be undertaken to identify the genera and possible range of species belonging to coralline red algae that encrust cobbles and boulders in the Upper

Pleistocene deposits at El Confital. Such a study necessarily entails the collection of samples for the making of thin sections transversely through crusts in order to identify features diagnostic at least on a genus level. Until such time, the best that can be said is that the present-day distribution of marine algae is widespread throughout the Canary Islands and keyed to habitats around individual islands in the Canary archipelago. Among the Corallinaceae known to be specific to Gran Canaria island, at least five genera are present, including *Hydrolithon*, *Lithophyullum*, *Lithoporella*, *Mesophylum*, and *Neogoniolithon*. Among these, *Lithophyllum* is the most diverse with four species locally attributed to that genus around the island [24]. From a paleoecological point of view, what is most telling about the boulder beds at El Confital at this stage of investigation is that they came to reside in shallow water within the upper photic zone consistent with the associated fossil fauna.

Overall, the combination of paleontological and physical evidence points to an open shelf setting on the margin of a small volcano around which a distinctly shallow-water biota thrived prior to interruption by storm events that eroded a series of parallel channels perpendicular to the strike of the paleoshore. These submarine gullies define the depositional space in which the Upper Pleistocene conglomerate was accommodated and preserved (Figure 3a,b).

Based on the application of two competing mathematical models that consider different dynamics [17,19], the estimated average height of storm waves that broke onto the south shore of La Isleta are remarkably similar in the range between 4.2 and 4.5 m (Table 2). However, the largest half-dozen boulders sampled from six study sites yield a much higher average estimated height of storm waves between 10.8 and 11.1 m (Table 2). Under any circumstances, such results would represent extremely large waves. These numbers represent clear outliers in the data, although based on boulders of extraordinary size and weight up to 2.25 metric tons. It may be that lesser waves were instrumental in gradually loosening the biggest basalt blocks from a joint-bound condition to a point where gravity slid them into nearby channels enlarged by multiple storm events. Rare hurricane events are more likely to have generated storm waves on the order of 6 to 8 m that impacted the Pleistocene rocky shore at El Confital. An entirely different set of equations is used to estimate the landward onrush of water due to tsunami events, but the complete absence of basalt boulders on the slopes of La Isleta (Figure 2) mitigates against this scenario.

**Table 2.** Summary data from Appendix A (Tables A1–A6) showing maximum bolder size and estimated weight compared to the average values for sampled boulders from each of the transects together with calculated values for wave heights estimated as necessary for boulder-beach mobility. Abbreviations: EAWH = estimated average wave height, EMWH = estimated maximum wave height.

| Confital Locality | Number of Samples | Average Boulder Volume (cm³) | Average Boulder Weight (kg) | EAWH (m) Nott [17] | EAWH (m) Pepe et al. [19] | Max. Boulder Volume (cm³) | Max. Boulder Weight (kg) | EMWH (m) Nott [17] | EMWH (m) Pepe et al. [19] |
|---|---|---|---|---|---|---|---|---|---|
| 1 | 30 | 28,065 | 79.7 | 4.5 | 3.9 | 212,173 | 602.6 | 7.4 | 12.7 |
| 2 | 28 | 9786 | 116.1 | 4.8 | 5.3 | 298,742 | 848.4 | 9.9 | 10.4 |
| 3 | 30 | 22,870 | 64.9 | 4.1 | 3.5 | 252,000 | 713.7 | 10 | 10.2 |
| 4 | 30 | 14,823 | 43.1 | 3.9 | 3.3 | 50,540 | 143.4 | 5.9 | 6.8 |
| 5 | 24 | 34,267 | 77.8 | 4.2 | 3.5 | 238,853 | 678.3 | 11.9 | 8.6 |
| 6 | 24 | 84,792 | 239.7 | 5.5 | 5.6 | 804,640 | 2258 | 21.5 | 16.1 |
| Average | 27.66 | 32,434 | 103.5 | 4.5 | 4.2 | 309,491 | 874.1 | 11.1 | 10.8 |

## 5.2. Inference from Historical Storms in the North Atlantic

Given their geographic location, archipelagos located north of Cabo Verde off the northwest coast of Africa are likely to be impacted by high-energy storms [25]. The Azores Archipelago is struck by high-energy storms with a frequency every seven years [26], causing several shipwrecks in the harbor of Ponta Delgada on São Miguel (Figure 7a). More recently, the passage of Hurricane Lorenzo in October 2019 caused the destruction

of several piers among the islands, as well as the near disappearance of Lajes das Flores harbor (Figure 7b). The Canary Islands are no exception. The high-energy events that affect the islands have caused considerable damage [12] and even fatalities (Figure 7c,d).

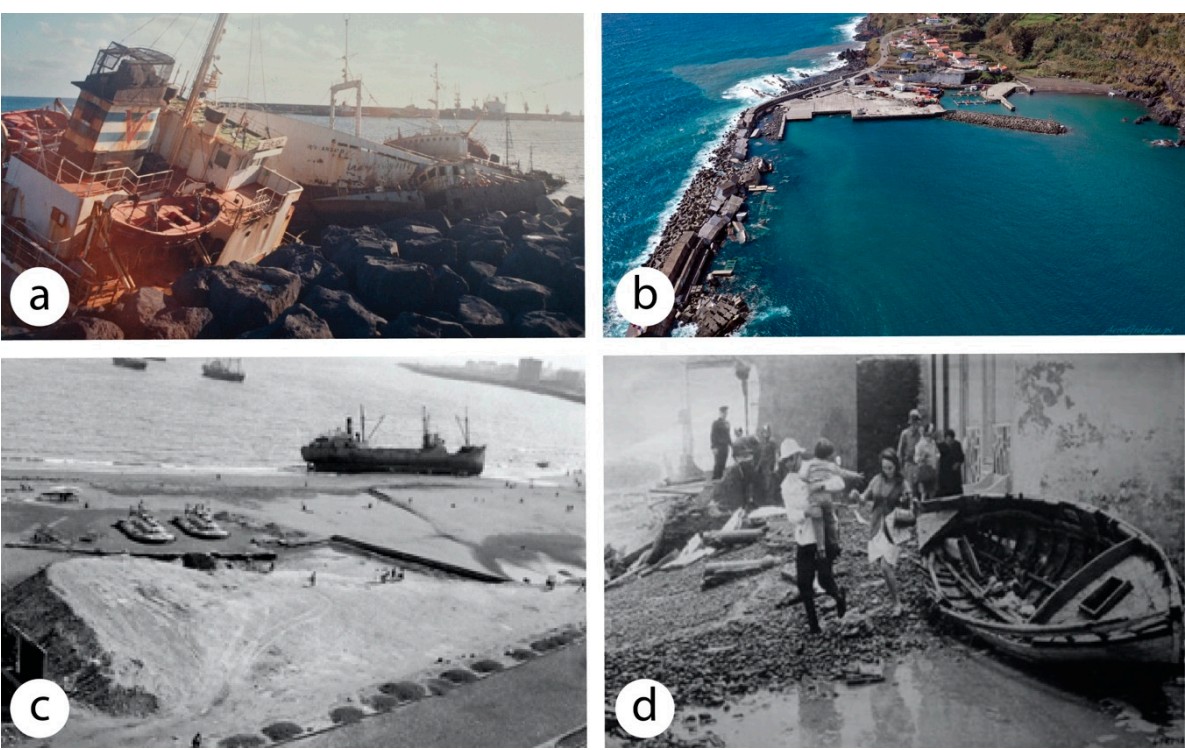

**Figure 7.** Evidence of destruction caused by storms in the Azorean and Canary islands; (**a**) Shipwrecks in Ponta Delgada harbor (São Miguel, Azores) after the December 1996 storm (photo by João Brum); (**b**) Destruction of the pier of Lajes das Flores (Flores Island, Azores) after the passage of Hurricane Lorenzo in October 2019 (Flores Island, Azores) after the passage of Hurricane Lorenzo in October 2019 (*Journal Açores* 9, 2019); (**c**) Shipwreck beached in Gran Canaria after the November 1968 storm (*Efemérides Meteorlógicas de Canárias*, 2018); (**d**) destruction caused by the April 1970 storm that hit Gran Canari (*Efemérides Meteorlógicas de Canárias*, 2018).

Historical records (Table 3) are scarce in terms of available wave-height information. However, in some cases inferences can be made: the events of 21 February 1966 (10 to 12 m), 23 November 1968 (10 m), and 21 April 1970 (8 m). These inferences suggest that storm waves can reach considerable values in the Canary Islands. Empirical models for wave modulation about the island of Gran Canaria indicate average wave height values between 5.22 and 5.58 m [27,28]. Such wave heights are compatible with the maximum average values estimated for boulders summarized for location 6 (Table 2) with averages based on Equation (1) [17] of 5.5 m and Equation (2) [19] of 5.6 m. The location of El Confital beach within the bay also is pertinent. The development of the volcanic edifice of La Isleta (Figure 1c) played an important role in protecting the area. Although consideration of the effect and action of waves before the Last Interglacial Epoch is possible, the formation of a large sandy isthmus since that time connected the La Isleta to the larger island of Gran Canaria, making the area even more sheltered from storm events. No direct evidence is observable at El Confital, but it is necessary to remember that the Canary Islands are subject to tsunami waves resulting from earthquakes like the wider regional event of 1755 that destroyed Lisbon, as well as volcanic flank collapse on the home islands.

**Table 3.** Coastal disturbances between 1755 and 2009 that affected Gran Canaria and islands elsewhere in the Canary Archipelago.

| Order | Date | Type Event | Locations | Scale of Destruction |
|:---:|:---:|:---:|:---:|:---|
| 1 | 25 January 1713 | major storm | Gran Canaria and Tenerife | Major destruction, overflow in Vega de Arucas |
| 2 | 1 November 1755 | tsunami | Tenerife, Gran Canaria, and Lanzarote | Destruction in coastal areas due to Lisbon earthquake |
| 3 | 6 January 1766 | major storm | Gran Canaria | Torrential rain; lahaar in Agüimes |
| 4 | 7–8 November 1826 | major storm | all islands | Damage throughout the Canaries and loss of life |
| 5 | 7 January 1856 | hurricane | Tenerife and El Hierro | 50 buildings destroyed including the harbor and pier; two fatalities |
| 6 | 13 February 1875 | major storm | Las Palmas de Gran Canaria | Damage to ships in the harbor, one fatality |
| 7 | 16 January 1895 | major storm | Gran Canaria | Overflow at E Confital, house destruction |
| 8 | 25–26 December 1898 | major storm | Gran Canaria | Damage to pier and docked ships |
| 9 | 3 March 1903 | major storm | Gran Canaria | No reported damage |
| 10 | 16 December 1903 | major storm | Gran Canaria | Raging sea; one fatality |
| 11 | 2 February 1904 | major storm | Gran Canaria | Disappearance of sand from El Confital |
| 12 | 28–29 October 1905 | major storm | Gran Canaria | Streets of Las Palmas flooded; one fatality |
| 13 | 17 January 1910 | major storm | Gran Canaria | Pier damage |
| 14 | 30 January 1910 | major storm | Gran Canaria | Overflow in Las Palmas; one fatality |
| 15 | 26–27 December 1910 | major storm | All islands | Las Palmas harbor closed to traffic |
| 16 | 1912 | major storm | Gran Canaria | Overflow in Las Palmas |
| 17 | 7–9 February 1912 | major storm | Gran Canaria | Homes in Las Canteras destroyed |
| 18 | 22 November 1913 | major storm | Gran Canaria | Homes in Las Canteras damaged |
| 19 | 26 November 1915 | major storm | Gran Canaria | Harbor closure |
| 20 | 19 November 1933 | major storm | Gran Canaria | Wave destruction; one fatality at El Confital |
| 21 | 12–13 December 1957 | major storm | Gran Canaria | Major destruction to Las Nieves harbor |
| 22 | 10 September 1961 | major storm | Gran Canaria | one fatality at Las Salinas de El Confital |
| 23 | 21 February 1966 | major storm | Gran Canaria | Wave heights between 10 and 12 m |
| 24 | 23–26 November 1968 | major storm | Tenerife and Gran Canaria | Wave height of 10 m in Tenerife; wind speed of 118 km/h in Gran Canaria |
| 25 | 21 April 1970 | major storm | Gran Canaria | Wave height of 8 m; wind speed over 90 km/h |
| 26 | 18 February 1971 | major storm | Gran Canaria | Damage in El Confital |
| 27 | February 1989 | major storm | Gran Canaria | Overflow; home damage; closure of Puerto de La Luz harbor |
| 28 | 28–29 November 2005 | major storm DELTA | all islands | Damage to harbors and beaches |
| 29 | 21 December 2009 | major storm | Gran Canaria | Overflow; precipitation levels of 130 mm/h; damage to homes |

### 5.3. Comparison with Coastal Boulder Deposits Elsewhere

Pleistocene and Holocene deposits formed by cobbles and boulders are widely distributed all around the world [3], but studies in coastal geomorphology seldom consider density as related to parent rock types when investigating the range of wave heights nec-

essary for their development as eroded boulders. Application of mathematical formulae such as Equation (1) from Nott [17] to estimate storm wave height has been applied previously to coastal boulder deposits throughout Mexico's Gulf of California, including those formed by limestone, rhyolite, and andesite clasts [4–6]. Extension of this work to include Equation (2) as derived from Pepe et al. [19] also has been applied to coastal basalt deposits in the Azores [1]. The variation in mass among these rock types ranges from 1.86 gm/cm$^3$ for limestone, 2.16 gm/cm$^3$ for the rhyolite, 2.55 gm/cm$^3$ for andesite, and 3.0 gm/cm$^3$ for basalt.

Among the largest Holocene boulders treated in these studies are those derived from rhyolite shores on the Gulf of California yielding mega-boulders with a calculated weight of 4.3 metric tons requiring an estimated wave height of 16.8 m to shift from the parent rocky shore [5]. In this particular case, the estimated wave height from storms in the Gulf of California is commensurate with the wave height formulated for the largest basalt boulder recorded anywhere on El Confital beach at study site 6 (Table 2). The recent history of hurricanes in the Gulf of California filmed under direct observation, confirms wave heights at least half that size against contemporary rhyolite sea cliffs [5].

The island of Santa Maria in the Azores has yielded a study comparing present-day and Upper Pleistocene basalt boulders [1] most applicable to the present study at El Confital beach in Gran Canaria. The largest Pleistocene boulder recorded in that study is smaller than many of the typical basalt boulders from El Confital, but still exemplified an estimated wave height of 8 m necessary for its emplacement. The largest boulder from a modern deposit on Santa Marine Island was calculated to require a wave height of 6.4 m for its emplacement [1].

Technically, hurricanes have a tropical to subtropical source dependent on high ocean-surface water temperatures and excessive air moisture [29]. However, major storms of hurricane intensity also occur in Arctic latitudes, where contemporary and Holocene boulder deposits occur along the coast of Norway. Small boulders formed by low-grade chromite ore with a density of 3.32 g/cm$^3$ are described from Holocene deposits on Norway's Leka Island that imply wave heights as much a 7 m for their emplacement [30].

### 5.4. Notes on the Geoheritage of El Confital Beach

El Confital beach takes its name from the former abundance of "confites" (candies in English) on the beach, a popular name given to rhodoliths throughout the Canary Islands for their white color and ball-like shape. Notably, the study area at El Conital beach is richly fossiliferous as known since the visit by Charles Lyell in 1854 to Gran Canaria island [31]. Historically, deposits with fossil rhodoliths, as well as other carbonates were exploited massively to manufacture lime, due to the lack of this resource in the Canary Islands. This industry has led to the loss of essential paleontological paleoecological, and taphonomical information. The outcrop at El Confital was chosen as a Geosite (site of geological interest, acronym LIG in Spanish) [32,33] and is included in the Inventory of Geosites of the Canary Islands, carried out by project LIGCANARIAS [11,34], due to its high scientific value that represents an area where different types of geological heritage are combined. Stratigraphic sequence is the central feature around which others are related including paleontology, sedimentology, and geomorphology [34–37]. The volcano-sedimentary sequence at El Confital reaches a maximum level of 200 m above sea level in which characteristics of the geological evolution of Gran Canaria island are represented.

As shown in this study, the significance of El Confital is magnified as an example of an accumulation zone of basaltic boulders of different sizes that denote high-energy events essential to understanding the impact of storms and hurricanes in the island groups of the North Atlantic. Apart from the materials belonging to the last interglacial maximum (MIS5e) described in this paper, El Confital includes a range of other features represented by submarine and subaerial basaltic deposits and hyaloclastites (peperites) together with marine sands and conglomerates, aeolian sand dunes, and colluvial deposits [35].

For all its value as a Geosite with high scientific, educational, and touristic value [32,33,35,36], El Confital beach is extremely fragile and vulnerable to human impact and climate change. Therefore, it remains necessary to adopt a management plan that ensures its regulatory protection in the short- and intermediate-term [37]. Although adjacent to the Bahia del Confital Special Conservation Area, Community Interest Area, as well as La Isleta Marine Area, and the Protected Landscape of La Isleta, it is urgent that the Geosite attain an effective geoconservation plan sanctioned by the regional government of the Canary Islands.

## 6. Conclusions

Study of the cobble-boulder deposits at Playa El Confital offers insights based on mathematical equations for estimation of Late Pleistocene wave heights from super-storms in the same region:

- Consolidated cobble-boulder deposits preserved in multiple samples from Upper Pleistocene strata exhibit evidence of major slumps from a rocky shoreline formed by basalt flows during a prior stage of development related to the small volcanic peninsula of La Isleta.
- Preserved in conglomerate deposits that are well cemented, the average estimated volume and weight of individual basalt boulders from a total of 166 samples suggest wave heights between 4.2 and 4.5 m responsible for their derivation from an adjacent stratiform and joint-bound body of parent rock. The largest basalt boulder from all six sample sites is estimated to weigh 2.25 metric tons and may have been moved by a wave of extraordinary height around 10 m. Alternately, smaller waves may have gradually loosened this block from its parent body until the force of gravity entrained it within the conglomerate.
- Often ellipsoidal to angular in shape but typically well rounded, the degree of wear to which individual boulders were subjected implies the action of multiple storm events that also gradually enlarged the size of the channels in which the conglomerates were entrained. Ellipsoidal shapes were governed by the spacing of vertical joints in the parent basalt flow.
- An associated marine biota consisting of diverse mollusks dates the conglomerates to an age consistent with Marine Isotope Stage 5e, equivalent to the Eemian Stage during the last interglacial epoch. The biota also includes rhodoliths formed by coralline red algae growing in spherical forms unattached to the seabed, as well as abundant evidence of platy red algae encrusted directly onto many boulders. Much the same biota lives the present-day embayment at El Confital and represents an inter-tidal to very shallow subtidal habitat.
- Historical records from major storm events that impacted the Canary and nearby Azorean islands confirm that wave heights in the range of those predicted by mathematical models for the erosion of the El Confital conglomerates are reasonable for erosion of all but perhaps the largest boulders entrained, therein.
- Given the importance of geoheritage at El Confital beach and the boulder deposits described in this paper, it remains necessary to implement an adequate management plan against human impact and climate change.

**Author Contributions:** M.E.J. initiated the project as a contribution to the Special Issue in the Journal of Marine Sciences and Engineering devoted to "Evaluation of Boulder Deposits Linked to Late Neogene Hurricane Events." I.G. and N.S. collected the extensive raw data on bolder dimensions at the study site. All co-authors contributed various parts of the text, with I.G. and N.S. contributing material on geologic background and characteristics of the conglomerate deposit. E.M.-G. was responsible for the paleontological and paleoecological description. M.E.J. and S.P.Á. compiled the statistics on wave heights. C.R. and C.S.M. researched and summarized existing historical records on the region's major storms. J.V. contributed the summary on geoheritage. All authors have read and agreed to the published version of the manuscript.

**Funding:** This research received funding from the Canarian Agency for Research, Innovation, and Society of Information (ACIISI) under the Government of the Canary Islands through the project ProID2017010159. C.S. Melo is the recipient of a PhD scholarship M.3a/F/100/2015 from FRCT/Açores 2020 from the Regional Fund for Science and Technology (FRCT) and benefitted from S.P.Á. "Projecto Exploratório" IF/00465 from the Foundation for Science and Technology (FCT). He also expresses appreciation for help from the FEDER through the Operational Program for Competitiveness Factors–COMPETE; by FCT under projects UID/BIA/50027/2013 and POCI-01-01-0145-FEDER-006821; DRCT1.1 project a/005/Function-C-/2016 (CIBIO-A) of the FRCT.

**Acknowledgments:** To be added following the review process.

**Conflicts of Interest:** The authors declare no conflict of interest.

## Appendix A

**Table A1.** Quantification of clast size, volume, and estimated weight from location 1 at the east end of Playa El Confital. The density of basalt at 2.84 g/cm$^2$ is applied uniformly in order to calculate wave height for each boulder on the basis of competing equations. Abbreviation: EWH = estimated wave height.

| Sample | Long Axis (cm) | Intermediate Axis (cm) | Short Axis (cm) | Volume (cm$^3$) | Adjust to 80% | Weight (kg) | EWH Nott [17] (m) | EWH Pepe et al. [19] (m) |
|---|---|---|---|---|---|---|---|---|
| 1 | 28.5 | 15 | 5.5 | 2351 | 1881 | 5.3 | 2.9 | 1.2 |
| 2 | 41 | 34 | 25 | 34,850 | 27,880 | 79.2 | 4.1 | 5.7 |
| 3 | 30 | 25 | 24 | 18,000 | 14,400 | 40.0 | 3.0 | 5.4 |
| 4 | 16 | 11 | 7 | 1232 | 986 | 2.8 | 1.6 | 1.6 |
| 5 | 29 | 21 | 13.2 | 8039 | 6431 | 18.2 | 2.9 | 3.0 |
| 6 | 44 | 19 | 11 | 9196 | 9357 | 26.6 | 4.4 | 2.5 |
| 7 | 28 | 25 | 9.5 | 6650 | 5320 | 15.0 | 2.8 | 2.2 |
| 8 | 58 | 53.5 | 17 | 52,751 | 42,201 | 119.9 | 5.8 | 3.8 |
| 9 | 54 | 46 | 25.5 | 63,342 | 50,674 | 143.9 | 5.4 | 5.8 |
| 10 | 68.5 | 34 | 20 | 46,580 | 37,264 | 105.8 | 6.9 | 4.5 |
| 11 | 37 | 23 | 22 | 18,722 | 14,978 | 42.5 | 3.7 | 5.0 |
| 12 | 35 | 34 | 18 | 21,420 | 17,136 | 48.7 | 3.5 | 4.1 |
| 13 | 24 | 16 | 8 | 3072 | 2458 | 7.0 | 2.4 | 1.8 |
| 14 | 14 | 10 | 7.5 | 1050 | 840 | 2.4 | 1.4 | 1.7 |
| 15 | 74 | 64 | 56 | 265,216 | 212,173 | 602.6 | 7.4 | 12.7 |
| 16 | 38 | 23 | 18 | 15,732 | 12,586 | 35.7 | 3.8 | 4.1 |
| 17 | 63 | 36 | 22 | 49,896 | 39,917 | 113.4 | 6.3 | 5.0 |
| 18 | 46 | 24 | 23 | 25,392 | 20,314 | 57.7 | 4.6 | 5.2 |
| 19 | 53 | 34 | 16 | 28,832 | 23,066 | 65.5 | 5.3 | 3.6 |
| 20 | 21 | 18 | 13 | 4914 | 3931 | 11.0 | 2.1 | 2.9 |
| 21 | 45 | 21 | 10 | 9450 | 7560 | 21.5 | 4.5 | 2.3 |
| 22 | 41 | 20.5 | 13 | 10,927 | 8741 | 24.8 | 4.1 | 2.9 |
| 23 | 75 | 45 | 15 | 50,625 | 40,500 | 115.0 | 7.5 | 3.4 |
| 24 | 45 | 41 | 24 | 44,280 | 35,424 | 100.6 | 4.5 | 5.4 |
| 25 | 72 | 36 | 23 | 59,616 | 47,693 | 135.4 | 7.2 | 5.2 |
| 26 | 71 | 34 | 18 | 43,452 | 34,762 | 98.7 | 7.1 | 4.1 |
| 27 | 88 | 59 | 23 | 119,416 | 95,533 | 271.3 | 8.8 | 5.2 |
| 28 | 43.5 | 26 | 13 | 14,703 | 11,762 | 33.4 | 4.4 | 2.9 |
| 29 | 23 | 18 | 10 | 4140 | 3312 | 9.4 | 2.3 | 2.3 |
| 30 | 40 | 33.5 | 12 | 16,080 | 12,864 | 36.5 | 4.0 | 2.7 |
| Average | 44.85 | 30 | 17.5 | 34,998 | 28,065 | 79.7 | 4.5 | 3.9 |

**Table A2.** Quantification of clast size, volume, and estimated weight from location 2 at the east end of Playa El Confital. The density of basalt at 2.84 g/cm² is applied uniformly in order to calculate wave height for each boulder on the basis of competing equations. Abbreviation: EWH = estimated wave height.

| Sample | Long Axis (cm) | Intermediate Axis (cm) | Short Axis (cm) | Volume (cm³) | Adjust to 80% | Weight (kg) | EWH Nott [17] (m) | EWH Pepe et al. [19] (m) |
|---|---|---|---|---|---|---|---|---|
| 1 | 99 | 82 | 46 | 373,428 | 298,742 | 848.4 | 9.9 | 10.4 |
| 2 | 71 | 48 | 27 | 92,016 | 73,613 | 209.1 | 7.1 | 6.1 |
| 3 | 27 | 19 | 10 | 5130 | 4104 | 11.7 | 2.7 | 2.3 |
| 4 | 55 | 27 | 25 | 37,125 | 29,700 | 10.5 | 5.5 | 5.7 |
| 5 | 33 | 16 | 8 | 4224 | 3379 | 9.6 | 3.3 | 1.8 |
| 6 | 46.5 | 45 | 43 | 89,978 | 71,982 | 204.4 | 4.7 | 9.7 |
| 7 | 62 | 24 | 24 | 35,712 | 28,570 | 81.1 | 6.2 | 5.4 |
| 8 | 56 | 30 | 20 | 33,600 | 26,880 | 76.3 | 5.6 | 4.5 |
| 9 | 58 | 51 | 35 | 103,530 | 82,824 | 235.2 | 5.8 | 7.9 |
| 10 | 19 | 15 | 7.5 | 2138 | 1710 | 4.9 | 1.9 | 1.7 |
| 11 | 47.5 | 42.5 | 37.5 | 75,703 | 60,563 | 172 | 4.8 | 8.5 |
| 12 | 46 | 30 | 54 | 74,520 | 59,616 | 169.3 | 4.6 | 12.2 |
| 13 | 64 | 42 | 36 | 96,768 | 77,414 | 219.9 | 6.4 | 8.2 |
| 14 | 47 | 29 | 24 | 32,712 | 26,170 | 74.3 | 4.7 | 5.4 |
| 15 | 27 | 18.5 | 7.5 | 3746 | 2997 | 8.5 | 2.7 | 1.7 |
| 16 | 48 | 35 | 23.5 | 39,480 | 31,584 | 89.7 | 4.8 | 5.3 |
| 17 | 77 | 46.5 | 25 | 89,513 | 71,610 | 203.4 | 7.7 | 5.7 |
| 18 | 42 | 41 | 20 | 34,440 | 27,552 | 78.2 | 4.2 | 4.5 |
| 19 | 33 | 31.5 | 13.5 | 14,033 | 11,227 | 31.9 | 3.3 | 3.1 |
| 20 | 43 | 40 | 27.5 | 47,300 | 37,840 | 107.5 | 4.3 | 6.2 |
| 21 | 36 | 34 | 14 | 17,136 | 13,709 | 38.9 | 3.6 | 3.2 |
| 22 | 19 | 17 | 13 | 4199 | 3359 | 9.5 | 1.9 | 2.9 |
| 23 | 69 | 24 | 21 | 34,776 | 27,821 | 79.0 | 6.9 | 4.8 |
| 24 | 56 | 31 | 24 | 41,664 | 33,331 | 94.7 | 5.6 | 5.4 |
| 25 | 30 | 17 | 15 | 7650 | 6120 | 17.4 | 3.0 | 3.4 |
| 26 | 31 | 26 | 18 | 14,508 | 11,606 | 33.0 | 3.1 | 4.1 |
| 27 | 53 | 32 | 16.5 | 27,984 | 22,387 | 63.6 | 5.3 | 3.7 |
| 28 | 43.5 | 31.5 | 22 | 30,146 | 24,116 | 68.5 | 4.4 | 5.0 |
| Average | 49 | 33 | 23.5 | 12,213 | 9786 | 116.1 | 4.8 | 5.3 |

**Table A3.** Quantification of clast size, volume, and estimated weight from location 3 at the east end of Playa El Confital. The density of basalt at 2.84 g/cm$^2$ is applied uniformly in order to calculate wave height for each boulder on the basis of competing equations. Abbreviation: EWH = estimated wave height.

| Sample | Long Axis (cm) | Intermediate Axis (cm) | Short Axis (cm) | Volume (cm$^3$) | Adjustto 80% | Weight (kg) | EWH Nott [17] (m) | EWH Pepe et al. [19] (m) |
|---|---|---|---|---|---|---|---|---|
| 1 | 53.5 | 26 | 24 | 33,384 | 26,707 | 75.8 | 5.4 | 5.4 |
| 2 | 45 | 29 | 22 | 28,710 | 22,968 | 65.2 | 4.5 | 5.0 |
| 3 | 31 | 26 | 17 | 13,702 | 10,962 | 31.1 | 3.1 | 3.8 |
| 4 | 50 | 30 | 23 | 34,500 | 27,600 | 78.3 | 5.0 | 5.2 |
| 5 | 33 | 25 | 17 | 14,025 | 11,220 | 31.9 | 3.3 | 3.8 |
| 6 | 100 | 70 | 45 | 315,000 | 252,000 | 713.7 | 10.0 | 10.2 |
| 7 | 22 | 18 | 17 | 6732 | 5386 | 15.3 | 2.2 | 3.8 |
| 8 | 25 | 13 | 11.5 | 3738 | 2990 | 8.5 | 2.5 | 2.6 |
| 9 | 27.5 | 12 | 10 | 3300 | 2640 | 7.5 | 2.8 | 2.3 |
| 10 | 52 | 23 | 14 | 16,744 | 13,395 | 38.0 | 5.2 | 3.2 |
| 11 | 17.5 | 20 | 20 | 7000 | 5600 | 15.9 | 1.8 | 4.5 |
| 12 | 44 | 11 | 11 | 5324 | 4259 | 12.1 | 4.4 | 2.5 |
| 13 | 47 | 19 | 19 | 16,967 | 13,574 | 38.6 | 4.7 | 4.3 |
| 14 | 35 | 11 | 11 | 11,235 | 3388 | 9.6 | 3.5 | 2.5 |
| 15 | 145 | 21 | 21 | 63,945 | 51,156 | 145.3 | 14.5 | 4.8 |
| 16 | 26 | 10 | 10 | 2600 | 2080 | 5.9 | 2.6 | 2.3 |
| 17 | 13.5 | 7 | 7 | 662 | 529 | 1.5 | 1.4 | 1.6 |
| 18 | 32 | 6 | 6 | 1152 | 922 | 2.6 | 3.2 | 1.4 |
| 19 | 52.5 | 18 | 18 | 17,010 | 13,608 | 38.6 | 5.3 | 4.1 |
| 20 | 26 | 8 | 8 | 1664 | 1331 | 3.8 | 2.6 | 1.8 |
| 21 | 30 | 16 | 16 | 7680 | 6144 | 17.4 | 3.0 | 3.6 |
| 22 | 23 | 18.5 | 18.5 | 7872 | 6297 | 17.5 | 2.3 | 4.2 |
| 23 | 120 | 44 | 44 | 232,320 | 185,856 | 527.8 | 12.0 | 10.0 |
| 24 | 28 | 14 | 14 | 5488 | 4390 | 12.5 | 2.8 | 3.2 |
| 25 | 40 | 16 | 16 | 10,240 | 8192 | 23.3 | 4.0 | 3.6 |
| 26 | 18 | 4.5 | 4.5 | 365 | 292 | 0.8 | 1.8 | 1.0 |
| 27 | 26 | 8 | 8 | 1664 | 1331 | 3.8 | 2.6 | 1.8 |
| 28 | 21 | 5 | 5 | 525 | 420 | 1.2 | 2.1 | 1.1 |
| 29 | 19 | 6.5 | 6.5 | 803 | 642 | 1.8 | 1.9 | 1.5 |
| 30 | 17 | 4 | 4 | 272 | 218 | 0.6 | 1.7 | 0.9 |
| Average | 79 | 18 | 15.5 | 28,821 | 22,870 | 64.9 | 4.1 | 3.5 |



**Table A4.** Quantification of clast size, volume, and estimated weight from location 4 on at the west end of Playa El Confital. The density of basalt at 2.84 g/cm² is applied uniformly in order to calculate wave height for each boulder on the basis of competing equations. Abbreviation: EWH = estimated wave height.

| Sample | Long Axis (cm) | Intermediate Axis (cm) | Short Axis (cm) | Volume (cm³) | Adjustto 80% | Weight (kg) | EWH Nott [17] (m) | EWH Pepe et al. [19] (m) |
|---|---|---|---|---|---|---|---|---|
| 1 | 55 | 38 | 18 | 37,620 | 30,096 | 85.5 | 5.5 | 4.1 |
| 2 | 52.5 | 42 | 21 | 46,305 | 37,044 | 105.2 | 5.3 | 4.8 |
| 3 | 50 | 25 | 22 | 27,500 | 22,000 | 62.48 | 5.0 | 5.0 |
| 4 | 59 | 53.5 | 20 | 63,130 | 50,504 | 143.4 | 5.9 | 4.5 |
| 5 | 35 | 19.5 | 22 | 15,015 | 12,012 | 34.1 | 3.5 | 5.0 |
| 6 | 31 | 21 | 17 | 11,067 | 8854 | 25.1 | 3.1 | 3.8 |
| 7 | 30 | 13 | 6 | 2340 | 1872 | 5.3 | 3.0 | 1.4 |
| 8 | 52 | 20 | 15 | 15,600 | 12,450 | 35.4 | 5.2 | 3.4 |
| 9 | 35 | 31.5 | 15 | 16,538 | 13,230 | 37.6 | 3.5 | 3.4 |
| 10 | 25 | 9 | 8.5 | 1913 | 1530 | 4.3 | 2.5 | 1.9 |
| 11 | 52 | 16 | 8 | 6656 | 5325 | 15.1 | 5.2 | 1.8 |
| 12 | 24 | 23 | 19 | 10,488 | 8390 | 23.8 | 2.4 | 4.3 |
| 13 | 28 | 21 | 7.5 | 4410 | 3528 | 10.0 | 2.8 | 1.7 |
| 14 | 42 | 20 | 10 | 8400 | 6720 | 19.1 | 4.2 | 2.3 |
| 15 | 42 | 29 | 21 | 25,578 | 20,462 | 58.1 | 4.2 | 4.8 |
| 16 | 32 | 19 | 10.5 | 6384 | 5107 | 14.5 | 3.2 | 2.4 |
| 17 | 41 | 37 | 9 | 13,653 | 10,922 | 31.0 | 4.1 | 2.0 |
| 18 | 23 | 12 | 11 | 3036 | 2429 | 6.9 | 2.3 | 2.5 |
| 19 | 38 | 29 | 19 | 20,938 | 16,750 | 47.6 | 3.8 | 4.3 |
| 20 | 22 | 14.5 | 9.5 | 3031 | 2424 | 6.9 | 2.2 | 2.2 |
| 21 | 25 | 9 | 5 | 1125 | 900 | 2.6 | 2.5 | 1.1 |
| 22 | 30 | 11 | 6.5 | 2145 | 1716 | 4.9 | 3.0 | 1.5 |
| 23 | 22 | 11 | 8 | 1936 | 1549 | 4.4 | 2.2 | 1.8 |
| 24 | 49 | 38 | 24 | 44,688 | 35,750 | 101.5 | 4.9 | 5.4 |
| 25 | 51 | 34 | 30 | 52,020 | 41,616 | 147.7 | 5.1 | 6.8 |
| 26 | 41 | 19.5 | 14 | 11,193 | 8954 | 25.4 | 4.1 | 3.2 |
| 27 | 31 | 21 | 7 | 4552 | 3646 | 10.4 | 3.1 | 1.6 |
| 28 | 56.5 | 51 | 20 | 57,630 | 46,104 | 130.9 | 5.7 | 4.5 |
| 29 | 41 | 26.5 | 14 | 15,211 | 12,169 | 34.6 | 4.1 | 3.2 |
| 30 | 55 | 43 | 20 | 25,800 | 20,640 | 58.6 | 4.3 | 4.5 |
| Average | 39 | 25 | 14.5 | 18,530 | 14,823 | 43.1 | 3.9 | 3.3 |

**Table A5.** Quantification of clast size, volume, and estimated weight from location 5 on at the west end of Playa El Confital. The density of basalt at 2.84 g/cm$^2$ is applied uniformly in order to calculate wave height for each boulder on the basis of competing equations. Abbreviation: EWH = estimated wave height.

| Sample | Long Axis (cm) | Intermediate Axis (cm) | Short Axis (cm) | Volume (cm$^3$) | Adjust to 80% | Weight (kg) | EWH Nott [17] (m) | EWH Pepe et al. [19] (m) |
|---|---|---|---|---|---|---|---|---|
| 1 | 57 | 28 | 22 | 35,112 | 28,090 | 79.8 | 5.7 | 5.0 |
| 2 | 76 | 37 | 20 | 56,240 | 44,992 | 127.8 | 7.6 | 4.5 |
| 3 | 38 | 28 | 27 | 28,728 | 22,982 | 65.3 | 3.8 | 6.1 |
| 4 | 46.5 | 30 | 20 | 27,900 | 22,320 | 63.4 | 4.7 | 4.5 |
| 5 | 46 | 34 | 22.5 | 35,190 | 28,152 | 80.0 | 4.6 | 5.1 |
| 6 | 61 | 30 | 21 | 38,430 | 30,744 | 87.3 | 6.1 | 4.8 |
| 7 | 45 | 38 | 11 | 18,810 | 15,048 | 42.7 | 4.5 | 2.5 |
| 8 | 59 | 43 | 38 | 96,406 | 77,125 | 219.0 | 5.9 | 8.6 |
| 9 | 114 | 97 | 27 | 298,566 | 238,853 | 678.3 | 11.9 | 6.1 |
| 10 | 81.5 | 42 | 25 | 85,575 | 68,460 | 194.4 | 8.2 | 5.7 |
| 11 | 43 | 28 | 17.5 | 21,070 | 16,856 | 47.9 | 4.3 | 4.0 |
| 12 | 41 | 28 | 13 | 14,924 | 11,939 | 33.9 | 4.1 | 2.9 |
| 13 | 39 | 33 | 17 | 21,879 | 17,503 | 49.7 | 3.9 | 3.8 |
| 14 | 30 | 25 | 7 | 5250 | 4200 | 11.9 | 3.0 | 1.6 |
| 15 | 22 | 11.5 | 6.5 | 1645 | 1316 | 3.7 | 2.2 | 1.5 |
| 16 | 27.5 | 14.5 | 4 | 1595 | 1276 | 3.6 | 2.8 | 0.9 |
| 17 | 20 | 8 | 5 | 800 | 640 | 1.8 | 2.0 | 1.1 |
| 18 | 32 | 27 | 27 | 23,328 | 18,662 | 53.0 | 3.2 | 6.1 |
| 19 | 19 | 10 | 7.5 | 1425 | 1140 | 3.2 | 1.9 | 1.7 |
| 20 | 17.5 | 16 | 9 | 2520 | 2016 | 5.7 | 1.8 | 2.0 |
| 21 | 27 | 17 | 6.5 | 2984 | 2387 | 6.8 | 2.7 | 1.5 |
| 22 | 22 | 12 | 6 | 1584 | 1267 | 3.6 | 2.2 | 1.4 |
| 23 | 12.5 | 7 | 4 | 350 | 280 | 0.8 | 1.3 | 0.9 |
| 24 | 19 | 11 | 10 | 2090 | 1672 | 4.7 | 1.9 | 2.3 |
| Average | 41.5 | 27 | 15.5 | 34,267 | 27,413 | 77.8 | 4.2 | 3.5 |

**Table A6.** Quantification of clast size, volume, and estimated weight from location 6 on at the west end of Playa El Confital. The density of basal at 2.84 g/cm$^2$ is applied uniformly in order to calculate wave height for each boulder on the basis of competing equations. Abbreviation: EWH = estimated wave height.

| Sample | Long Axis (cm) | Intermediate Axis (cm) | Short Axis (cm) | Volume (cm$^3$) | Adjust to 80% | Weight (kg) | EWH Nott [17] (m) | EWH Pepe et al. [19] (m) |
|---|---|---|---|---|---|---|---|---|
| 1 | 58 | 54 | 34 | 106,488 | 85,190 | 242.0 | 5.8 | 7.7 |
| 2 | 214 | 94 | 50 | 1,005,800 | 804,640 | 2258.0 | 21.5 | 11.3 |
| 3 | 84 | 83 | 71 | 495,012 | 396,010 | 1124.6 | 8.4 | 16.1 |
| 4 | 47 | 33 | 26 | 40,326 | 32,261 | 91.6 | 4.7 | 5.9 |
| 5 | 33 | 17 | 13 | 7293 | 5834 | 16.6 | 3.3 | 2.9 |
| 6 | 50.5 | 49 | 28 | 69,286 | 55,429 | 157.4 | 5.1 | 6.3 |
| 7 | 70.5 | 52 | 32 | 117,312 | 93,850 | 266.5 | 7.1 | 7.2 |
| 8 | 85 | 44 | 32 | 119,680 | 95,744 | 271.9 | 8.5 | 7.2 |
| 9 | 40 | 31 | 13 | 16,120 | 12,896 | 36.6 | 4.0 | 2.9 |
| 10 | 46.5 | 24 | 16 | 17,856 | 14,285 | 40.6 | 4.7 | 3.6 |
| 11 | 34.5 | 27.5 | 17.5 | 16,603 | 13,283 | 37.7 | 3.5 | 4.0 |
| 12 | 31 | 24 | 18 | 13,392 | 10,714 | 30.4 | 3.1 | 4.1 |
| 13 | 40.5 | 24 | 23 | 22,356 | 17,885 | 50.6 | 4.1 | 5.2 |
| 14 | 57 | 41 | 27 | 63,099 | 50,479 | 143.3 | 5.7 | 6.1 |
| 15 | 65 | 60 | 41 | 159,900 | 127,920 | 363.3 | 6.5 | 9.3 |
| 16 | 41 | 38 | 25 | 38,950 | 31,160 | 88.5 | 4.1 | 5.7 |
| 17 | 33 | 21 | 16 | 11,088 | 8870 | 25.2 | 3.3 | 3.6 |
| 18 | 92 | 61 | 26 | 145,912 | 116,730 | 331.5 | 9.2 | 5.9 |
| 19 | 27 | 11 | 11 | 3267 | 2614 | 7.4 | 2.7 | 2.5 |
| 20 | 44 | 41 | 7 | 12,628 | 10,102 | 28.7 | 4.4 | 1.6 |
| 21 | 13 | 9.5 | 5 | 618 | 494 | 1.4 | 1.3 | 1.1 |
| 22 | 30 | 30 | 29 | 26,100 | 20,880 | 59.3 | 3.0 | 6.6 |
| 23 | 41 | 27 | 18 | 19,926 | 15,941 | 45.3 | 4.1 | 4.1 |
| 24 | 41 | 24 | 15 | 14,760 | 11,808 | 33.5 | 4.1 | 3.4 |
| Average | 55 | 38 | 25 | 105,991 | 84,792 | 239.7 | 5.5 | 5.6 |

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
