# Peer review of "Late Pleistocene Boulder Slumps Eroded from a Basalt Shoreline at El Confital Beach on Gran Canaria (Canary Islands, Spain)"

_jmse, doi:10.3390/jmse9020138_

Round 1

Reviewer 1 Report

Galindo et al. report the characters of Late-Pleistocene boulders at El Confital Beach and display the geological significance of these boulders. Hurricane intensity was also inferred from their observations. I think these results are new and significant in geology, which provides a unique window for extreme events in the past. My comments are listed as follows:

1) The map of the study is too simple. How about changes in elevation? How about the visual difference between two regions of collecting samples? Too much green colors, and it is difficult to identify them, labelled some in words?

2) The geological background is not enough. The authors discussed a lot about waves and hurricanes, and listed some characters about it in discussion part and tables. There should be some in the second part, some characters about it, some modern observations, or meteorological records.

3) The shape and size of boulders. Are there any priority directions for these boulders in stratum? How about rose map?

4) Many figures miss legends and/or scales.

5) It seems that many words in section 5.1 are not closely related.

6) Is it possible to date these boulders in future?

7) Some words are not easy to follow, and some typos needs edits.

Reviewer 2 Report

This really interesting contribution on a hotly-debated subject. It presents outcomes from a state-of-the-art project. The paper is informative, well-illustrated, and bearing representative list of references. I see only several small issues for revision.

  • Abstract: please, avoid justifying your writing to a project. The words 'paper', 'study', etc. are more suitable.
  • The section 2 should be extended to provide more details and citations. I also encourage to separate geographical and geological information (give these, please, in different paragraphs).
  • Figure 1: what is the source of this map?
  • The information from 4.1 is more relevant to the section 2.
  • Lines 170-171: if boulders are >2 m, is it sensible to call them with the megaclast terminology (several terminological systems exist). Anyway, I do not insist on doing this, if the authors judge this a marginal issue.
  • Section 5.4: please, extend this section a bit, starting with explanation of the urgency of geoheritage studies in general and some notes on the world coastal geoheritage, and then going to the geoheritage in Spain and the Canary Islands. All these considerations should include citations to the literature.
  • Conclusions: add the concluding remarks from Discussion, including geoheritage section.
  • In some cases, you spell 'breach'. Do you mean 'beach'? Please, check.
  • Please, avoid sections consisting of a single paragraph: each section should include, at least, two paragraphs.
  • As for the taxa of extant and extinct organisms, some readers would prefer see these fully, i.e., with the authors and years (look at this critical note about the modern journals: https://www.sciencedirect.com/science/article/abs/pii/S001678782030033X ). However, I leave to the authors to decide whether to supply this information or not.
